# The Experiences of People with Dementia and Informal Carers Related to the Closure of Social and Medical Services in Poland during the COVID-19 Pandemic—A Qualitative Study

**DOI:** 10.3390/healthcare9121677

**Published:** 2021-12-03

**Authors:** Maria Maćkowiak, Adrianna Senczyszyn, Katarzyna Lion, Elżbieta Trypka, Monika Małecka, Marta Ciułkowicz, Justyna Mazurek, Roksana Świderska, Clarissa Giebel, Mark Gabbay, Joanna Rymaszewska, Dorota Szcześniak

**Affiliations:** 1Department of Psychiatry, Wroclaw Medical University, 50-367 Wroclaw, Poland; adrianna.senczyszyn@student.umw.edu.pl (A.S.); elzbieta.trypka@umw.edu.pl (E.T.); monika.malecka@student.umw.edu.pl (M.M.); marta.ciulkowicz@student.umw.edu.pl (M.C.); joanna.rymaszewska@umw.edu.pl (J.R.); dorota.szczesniak@umw.edu.pl (D.S.); 2Menzies Health Institute Queensland, Griffith University, Brisbane, QLD 4111, Australia; k.lion@griffith.edu.au; 3University Rehabilitation Centre, Wroclaw Medical University, 50-556 Wroclaw, Poland; justyna.mazurek@umw.edu.pl; 4School of Medicine, University of Liverpool, Liverpool L69 3GE, UK; roksanaswiderska92@gmail.com; 5Department of Primary Care and Mental Health, University of Liverpool, Liverpool L69 3GF, UK; clarissa.giebel@liverpool.ac.uk (C.G.); mbg@liverpool.ac.uk (M.G.); 6NIHR ARC NWC, Liverpool L69 3GL, UK

**Keywords:** dementia, COVID-19, social support, medical support, people with dementia, informal carers

## Abstract

Older people with dementia are particularly at risk of COVID-19; however, relatively little is known about the indirect impact of the pandemic on the lives of those living with, and/or caring for someone with, dementia. The aim of this study was to investigate the experiences of people with dementia and informal carers during the closure of available social and medical services in Poland during the COVID-19 pandemic. A qualitative thematic analysis of semi-structured interviews with people with dementia (*n* = 5) and informal carers (*n* = 21) was performed between June and August 2020 after the first wave of COVID-19 in Poland. Three overarching themes were identified: (1) care re-organization; (2) psychological responses; (3) emerging needs. The factor underlying all these elements was reliance on other people. Social support and engagement are vital to the ongoing health and well-being of people living with dementia and their informal carers. Services need to be strengthened to provide ongoing provision to those living with dementia to reach pre-pandemic levels, if not better. Within the post-pandemic environment, people with dementia and their informal carers need reassurance that they can rely on external institutional and social support able to meet their needs.

## 1. Introduction

As the protracted COVID-19 pandemic waxes and wanes, the needs of people living with dementia and their informal carers remain largely unaddressed and poorly understood [1,2,3]. These key therapeutic needs are primarily associated with daily activity, companionship and emotional support which significantly influence the quality of life of a person with dementia and influence the course of the disease [4]. Coronavirus has particularly threatened the physical health of people with dementia who might be more prone to infection, usually through older age, comorbidities or frailty [5], leading to moves to protect and isolate them from exposure which has had significant unintended adverse consequences for their dementia and emotional well-being [1,6,7,8].

Social health is usefully defined as the opportunity to fulfill one’s potential through individual resources and a stimulating social environment [9,10,11]. It has three pillars, which are: (1) the ability of people with dementia to fulfil their potential and obligations, (2) the ability to manage life with some degree of independence despite the disease and (3) participation in meaningful social activities and interactions [12]. The pandemic restrictions and recommendations for maintaining physical distance deprived people with dementia of a stimulating social environment, the second pillar of social health. The imposed social isolation withdrew people from interpersonal contacts which, until now, were opportunities to fulfill their social roles and assisted in symptom-related constraints. The balance between the capabilities and limitations maintained by a supportive environment was decisive for overall health [1,11,13].

The majority of people with dementia in Poland live within their local communities and are cared for by their families, who may live with, or be close to, them [14,15]. Until the outbreak of the pandemic, social engagement of people with dementia was provided both formally and informally. Some people with dementia could benefit from institutional support, i.e., provided by day care facilities (daily activities, cognitive stimulation and company) [16,17]. However, due to the insufficient institutional post-diagnostic support in Poland, a large proportion of people with dementia are not supported by formal care services, but only by their relatives [14,18]. Their social involvement is based on contacts with family and neighbors, and on daily activities. During the first wave of the pandemic, the available support options were suspended or significantly reduced [18,19]. Day care facilities were legally suspended [20] and contacts with relatives, in accordance with sanitary recommendations, were kept to a minimum [6,21].

Emerging evidence suggests that constraints in social participation of a person with dementia and limited access to social services have affected the entire family system, particularly the primary informal carer [22,23], who is usually left alone with their caring responsibilities due to the reduction in all forms of support they previously obtained [24]. Research conducted since the outbreak of the COVID-19 pandemic [25] appears to reinforce the finding from pre-pandemic research on informal carers’ burden [26]. Already a few years ago, researchers identified informal carers of people with dementia as the invisible second patients [27] who experience a high level of burden, organizational strain, psychological and physical morbidity and social isolation [27,28]. In the face of an ongoing pandemic, previously recognized issues seem to be escalated [25]. The frequently high levels of stress associated with caring for a person living with dementia [28] have been intensified even further by the pandemic and have negatively influenced carers’ overall psychological well-being [8,22,29]. For informal carers who are professionally active, switching to remote work became their only means to provide their relatives with the necessary care, imposing additional caring responsibilities upon them [30]. Those with lower levels of resilience were even more liable to experience increased mental health symptoms such as anxiety and depression [31]. Although evidence highlights the negative consequences of the pandemic on the informal carer’s role, for others, it has reinforced their existing isolation from the mainstream society as a consequence of their caregiving role [32,33]. For some, the pandemic, which has resulted in spending more time with their relatives, has potentially enriched their relationship [30,34]. Previous studies [29,32,35] provided insight into the impacts of pandemics on selected aspects of the psychological functioning of people with dementia and their informal carers, but few related these phenomena directly to the issue of using care services during pandemics [36,37], and the number of studies conducted in Poland was negligible [19].

The aim of this study was to investigate the experiences of people with dementia and informal carers related to the closure of available social and medical services in Poland during the COVID-19 pandemic. The evidence from this national study provides further insights into the global effects of the pandemic on those affected by dementia.

## 2. Methods

### 2.1. Study Design

We adopted a qualitative design to investigate the experiences of people with dementia and informal carers related to the closure of available social and medical services in Poland during the COVID-19 pandemic. We chose a thematic analysis as a convenient and in-depth method for exploring individual experiences related to the actual social context [38]. Due to the limited data on the presented topic, especially in the studied area, exploration of the personal insights can reveal unforeseen issues [38,39]. Consequently, this has potential to generate new hypotheses and set directions for further research in the field [40].

### 2.2. Recruitment Process

Informal carers aged 18+ who were caring for a person with dementia and people with dementia (with capacity to consent and comprehend the questions stated by a psychiatrist in a clinical assessment) accessing social care and/or social support services were enrolled in the study. Social care and social support services included: day care facilities, support groups, home rehabilitation, befriending services, visiting social worker, meal delivery and home care. The participants were recruited via outpatient memory clinics and day care facilities in the city of Wroclaw, Poland (the 4th largest urban agglomeration in Poland, with an estimated population of over 64,000) [41].

### 2.3. Data Collection

Informal carers and people with dementia who were interested in taking part in the study were contacted by telephone and invited to participate in a semi-structured telephone interview, and consented if still willing once the study was fully explained to them. The interviews were conducted by psychologists and psychiatrists between June and August 2020, after the first wave of COVID-19, when pandemic restrictions in Poland were eased.

A semi-structured interview topic guide was adapted from a parallel study conducted by Giebel et al. (2020) during April 2020 [36] and was translated according to the WHO translation protocol [42], which included a forward translation, a panel of experts, back translation, pre-testing and creation of the final version [43]. This included questions about service use before and since the COVID-19 outbreak. Each interview was audio recorded, and the mean interview length was *M* = 18 min (*SD* = 6 min 14 s).

### 2.4. Data Analysis

Recorded interviews were transcribed into verbatim scripts by the research team members experienced in preparing materials for qualitative analyses. Prior to analysis, all transcripts were anonymized and proofread. Reviewing the transcripts for correctness was, itself, a stage of the authors’ familiarization with the data. Four researchers analyzed the data—three psychologists and one psychiatry resident. The transcripts were analyzed by applying thematic analysis—both inductive and deductive [44]—with regard to the main analytical question: “What are the experiences of people with dementia and informal carers related to the closure of available social and medical services in Poland during the COVID-19 pandemic?” Analyses were conducted separately for informal carers and for people with dementia.

At the first stage, the most information-rich transcripts of interviews with informal carers and people with dementia (one from each group, with the most extensive responses of the participants) were analyzed independently by two researchers who developed initial codes (inductive analysis). In the discussion between researchers, the codes were made uniform and gathered into a codebook. Based on this jointly developed codebook, the remaining material was analyzed by one of the researchers (deductive analysis). If there was an extract of data relevant to the analytic question that did not match any of the codes classified primarily in the codebook, the new code was assigned and added to the codebook. The coding of transcripts was performed using NVivo software (QSR International, Gambit COiS Sp.z.o.o. Karków, Poland, 2020). The researchers involved in the analysis during the brainstorm session grouped the individual codes into the potential themes and sub-themes. Afterwards, the researchers formulated themes from the coded data with all authors, familiarized themselves with the transcripts and codes and discussed the relevance and naming of the formulated themes and sub-themes until agreement on them was reached.

The interviews, transcription and analysis were conducted in Polish. The results are reported in English. All the quotes cited were subject to back translation to ensure the correct transfer of meaning between languages.

## 3. Results

### 3.1. Participant Demographics

We conducted 26 semi-structured interviews with informal carers (*n* = 21) and people with dementia (*n* = 5). Informal carers and people with dementia were recruited separately and they were not dyads. The majority of the sample was female (17.65%), and the majority of informal carers (62%) were living with a person with dementia and were adult children (71%). The mean age of people with dementia was 81 (*SD* = 6.6; *Ra* = (70–87)), with their average period of education being 13 years (*SD* = 1.64; *Ra* = (9–12)). Participant demographics are summarized in Table 1.

### 3.2. Qualitative Themes

Thematic analysis resulted in three main themes with several sub-themes. Themes were common between informal carers and people with dementia: (1) care re-organization; (2) psychological response; (3) emerging needs. Arrangement of the themes and sub-themes is illustrated in Figure 1. The most representative quotes are presented in the text. More examples are available in Appendix A.

### 3.3. Theme 1: Care Re-Organization

Restrictions imposed due to the pandemic outbreak caused significant changes in the organization of care for people with dementia. To avoid contact with potential virus carriers, people with dementia withdrew from most of their external daily activities, including attendance at day care facilities. Consequently, informal carers were tasked with providing intensified care to their relatives. The need to adapt to these changes resulted in individual psychological reactions.

#### 3.3.1. Shutdown or Restrictions on Operation of Day Care Facilities

Differences in care organization during the pandemic were especially noticeable to those participants who were beneficiaries of social care facilities before. Day-to-day activities in a peer group and professional support turned out to be priceless and indispensable. Lack of a regular schedule, hours spent alone and restrictions on their own home were distressing for people with dementia and also influenced informal carers. The latter became unable to divide their care responsibilities and were worried that they would not provide enough cognitive, social and physical stimulation to their relatives.

*“What I felt when they closed the day care facility? It was stress. And it still is. We had to organize something that worked well again. I didn’t know when it would end, how we should work with mum, so that what was achieved in Senior Plus [daily care home] wouldn’t be wasted. We share the care with our sister, but it can be difficult.”* [Female Informal Carer, 65, Daughter, Interview 7]

*“As we were closed and not able to meet with others, it just felt like prison. It was hard to handle.”* [Female, Person with dementia, 82, Interview 1]

#### 3.3.2. Access to Medical Help

All participants reported that COVID-19 had become a priority for the entire healthcare system, with other medical conditions sidelined. As a result, access to medical care was often challenging, especially for people with dementia and other comorbidities. Interviewees pointed to the barriers in accessing teleconsultations, which included waiting queues. Older adults highlighted the importance of a doctor’s physical examination. The lack of it caused uncertainty about diagnosis and a sense of a decrease in the quality of care. Nevertheless, some participants were satisfied with telemedical solutions. This was largely dependent on the dedication of medical personnel in each facility.

*“I haven’t had any revolution in my life because there are telephones and I could always call for help and ask. We also have clinics here, the health care team, which is very dynamic. They are even so determined to help patients that they took us in during a pandemic when my husband’s blood sugar was too high.”* [Female Informal Carer, 77, Wife, Interview 19]

*“I wish the access to medical care was better.”* [Female, Person with dementia, 76, Interview 4]

#### 3.3.3. Burden of Care

Facing service closures and for fear of exposing older adults to the virus, many informal carers decided to strictly limit the number of contacts for their relatives. The duty of care usually fell on the closest person (children or a spouse) who could not count on any replacement or respite care resources. Participants underlined that it was a substantial organizational and emotional burden.

*“If the carer has an institution where parents spend their time and suddenly this institution closes, then it has a negative effect on their organization of life, right?”* [Female Informal Care, 58, Daughter, Interview 16]

*“My daughter always said: ‘Mum, you are going out-don’t go out!’ So as to avoid this event.”* [Female, Person with dementia, 70, Interview 5]

### 3.4. Theme 2: Psychological Response

Changes in the organization of everyday life and unpredictable situations initiated the process of adaptation to changed life circumstances. This process triggered different emotional reactions in informal carers and people with dementia and resulted in various attitudes and individual consequences—from difficulties in adapting to the situation to implementing positive coping strategies.

#### 3.4.1. Uncertainty and Anxiety

Participants experienced many negative emotions evoked by the pandemic crisis. Fear of being infected combined with the additional caring duties or self-isolation resulted in an apparent decline in respondents’ mental health. Informal carers reported increased levels of distress related mostly to their concerns about their relative with dementia. For people with dementia, pressure to stay at home increased their sense of helplessness, lack of agency and anxiety.

*“It was very sad. I was lying on my bed and staring at the wall. Every day. How many people died and where? It was very sad. It lasted for so long.”* [Female Informal Carer, 58, Daughter in-law, Interview 10]

*“**Well, I got a little stressed [with the lockdown], because it is a kind of burden. If I need something, some help, there is no way to get it. If I call a doctor, it is also impossible to get to him. And it’s even hard to get a prescription.”* [Female, Person with dementia, 76, Interview 4]

#### 3.4.2. Social Isolation

Physical distancing not only impacted the organizational aspects of life but also brought on an emotional void and intensified the sense of loneliness, which particularly affected older adults. Informal carers felt powerless in the face of the adverse consequences of social isolation on their relatives. They apprehensively observed a decline in the overall functioning of people with dementia including cognitive skills, emotional well-being and physical functioning. They were concerned whether people with dementia would ever be able to return to the pre-pandemic levels of mental and physical health.

*Literally, very soon after the day care facility was closed, my mother started to deteriorate in her health, especially the mental one. Her behaviours started to change, a lot of problems grew and for me it was a very big problem. I had to hire a private carer very quickly.”* [Female Informal Carer, 58, Daughter, Interview 1]

*“I am used to being lonely. Even before that virus I lived a life of a lonely person as I had been caring for my wife for 25 years.”* [Male, Person with dementia, 87, Interview 2]

#### 3.4.3. Adaptation and Coping

Although the COVID-19 pandemic negatively influenced many aspects of functioning, some people were able to adapt to the new circumstances. Their capacity to maintain a mental balance depended on the adoption of positive coping strategies, both emotion focused and problem focused. The extent to which people with dementia adjusted to the sanitary recommendations and accepted imposed requirements was notable.

*“If it weren’t for the optimism, we couldn’t deal with such problems.”* [Female Carer, 75, Wife, Interview 9]

*“I don’t mind wearing masks if it’s required.”* [Male, Person with dementia, 87, Interview 2]

### 3.5. Theme 3: Emerging Needs

Pandemic-related changes and the process of adjusting to the new reality revealed important needs for people with dementia and informal carers. Participants indicated what was helpful in coping with the pandemic situation, and what was missing but would be valuable.

#### 3.5.1. Institutional Support

Respondents claimed that institutional support during the pandemic was insufficient and often did not meet their expectations. Help from the welfare office or personnel of day care facilities was either absent or not adequate. Most of the informal carers had to organize everything themselves. Those who received some institutional support highly appreciated it.

*“Regarding the psychosocial interventions, unfortunately it was very poor. It is true that from time to time, personnel [of daily care home] sent some links to on-line exercises, but my mother cannot use it. She can’t handle something like that, so it was unfortunately useless for her.”* [Male, Informal Carer, 58, Son, Interview 15]

#### 3.5.2. Social Support

Irrespective of institutional support, participants emphasized the indispensable importance of social support. Regardless of whether it was family, friends or neighbors, any support from others prevented people with dementia and informal carers from feeling left alone. Perception of other people’s willingness to help was a vital factor influencing the experience of the pandemic period, notably in reducing negative emotions.

*“Of course, we are supported by friends and family. It would be hard without it. Or even hopeless.”* [Female Informal Carer, 77, Wife, Interview 19]

*“I have had thoughts how it’s going to be like and if I can handle it. But what turns out is that there are people who remember me. And they help. And this is very important for me. And that’s what they tell me: if you needed help, we would help you, just tell us.”* [Female, Person with dementia, 75, Interview 3]

#### 3.5.3. Remote Contacts

Informal carers appreciated online forms of communication. For them, it was a useful medium of connecting with people in a time of social distancing. The Internet (social media, online support groups) allowed them to receive information about providing dementia support and reduced loneliness. However, remote forms of communication did not satisfy the needs of people with dementia. Engaging in remote communication was considered too technically challenging and did not offer emotional closeness to another person.

*“It [on-line support for carers via communicators such as Zoom, Skype] is important. Because if there is no direct possibility, then you just have to look for a solution and undoubtedly some social media or some platforms that allow you to contact, (and see another person, because it is the human face that has the power), it is important. It is a form that may not be perfect, but it is good enough to be used.”* [Female, Informal Carer, 60, Daughter, Interview 17]

*“These teleconsultations… well, maybe they don’t quite meet my expectations. Actually, mine and my husband’s. Because we have a cold, for example, we want the doctor to see us, but to auscultate the throat, to take care of the patient in such a professional way.”* [Female, Person with dementia, 70, Interview 5]

## 4. Discussion

This study highlights the experiences of people with dementia and informal carers related to the pandemic’s public health restrictions and adds to a developing evidence base about the detrimental impacts of COVID-19 and associated restrictions on the lives of some of the most vulnerable members of society and their informal carers [36,45,46,47,48]. The results reveal various aspects of the pandemic’s consequences on people’s lives and changes in their care needs. They emphasize that while individuals may have individual resources, social contacts are critical for overall health and well-being in dementia.

Coronavirus restrictions suspended the functioning of the majority of care services available in Poland [49]. The results from this study show that both people with dementia and informal carers had to re-plan their care and switch to remote ways of using social and medical services. The sudden changes created a variety of psychological responses, such as emotional distress or helplessness. This started the adaptation to the critical situation [50]. The challenges posed by the pandemic additionally revealed support needs—both institutional and social—which were helpful for better adaptation to life with dementia under the pandemic regime [51].

The actual changes in the functioning of the health and social care systems reported in this study are comparable to actions taken by governments of countries struggling with the novel coronavirus around the world [52]. Prioritizing reductions in viral transmission and prioritizing care for those severely ill with COVID-19 have left dementia care needs behind. This has negatively affected the mental and social well-being of people with dementia and those who support them [6,19,24].

In Poland, even before the pandemic, the dementia care system was not well organized and the major burden of care issues were left to informal carers [14,15], which is frequently the case across the globe [28,53]. As shown in our results, pandemic restrictions added anti-virus protection tasks to daily caring duties. Despite the extra effort put into caring, they considered the support offered to not be sufficiently stimulating for people with dementia. Consequently, informal carers reported elevated physical and psychological strain. This adverse trend, according to the observations conducted thus far, has become a common experience of informal carers around the world, regardless of the care system operating in a given country [1,19,22,35,47,54]. Pandemic-related changes in the caring responsibilities have lowered informal carers’ overall well-being, both mental and physical [33], causing symptoms such as sleep disturbances, anxiety or depression [55]. Moreover, it has increased the risk of carer burnout [47]. The intensity of these psychological reactions largely depends on intrapersonal factors such as their appraisal of the pandemic risks and consequences and their ability to adopt coping strategies [50]. Thus, the pandemic has further highlighted the need for better support infrastructure for informal carers. It is not feasible for informal carers to conduct most of the care for a person with dementia, and pandemic-related learning needs to include improved access to post-diagnostic dementia care, enabling more respite for informal carers.

Among people with dementia, we observed a burden resulting from limited interactions with the environment and feeling that one’s hands are tied. Increased dependence on others with simultaneous withdrawal from the social environment disrupted the elements underlying social health [1]. Despite the preservation of personal resources such as the ability to adapt and cope with the new situation, people with dementia were deprived of the most important factors against the faster cognitive deterioration and decline in their overall well-being [1,6,10,12]. This situation corresponds to Talbot and Briggs’ “shrinking world” theory [45] relating to the experience after receiving a dementia diagnosis [56]. COVID-19 has escalated the dominant sensation following a dementia diagnosis that the world of a person is narrowing. It has deprived people with dementia of the social foothold and disrupted their regular activities, posing another loss [45]. On the other hand, however, being restricted to home may have offered relief from outside sources of anxiety and could prevent people from engaging with everyday activities [57]. As a result, people with dementia were deprived of the factors helping to maintain their independence such as autonomy in dealing with everyday duties or not being treated by others in an overprotective manner [4,58,59,60]. Uncertainty around the coronavirus has added to the loss of the sense of control [61] and adversely affected the overall well-being of people with dementia [62,63,64].

Lack of environmental stimulation caused by pandemic restrictions turned out to be a considerable accelerator of decline in emotional and cognitive functioning [36,65]. Research suggests a wide range of specific neuropsychiatric symptoms which are mediated by this form of imposed, elongated social isolation. Among them, there are mood alterations, apathy, anxiety, reduced motor activity and appetite, circadian rhythm changes and psychotic symptoms [62,64,66,67,68,69]. In our study, it was the informal carers who reported changes in the functioning of people with dementia. They observed most of those neuropsychiatric symptoms noted elsewhere [62,64,66,70] and attributed them to the imposed social isolation and closures of social services. In the face of these adverse psychological and psychiatric consequences, it is important to note that both people with dementia and informal carers are exposed to increased social isolation regardless of the pandemic restrictions [32,47,64]. Ongoing crises only heighten the risk of the negative effects of loneliness, thereby highlighting the importance of social support in dementia care in general [19,62,71,72].

Despite the negative psychosocial consequences, our results show that people with dementia did not question the public health guidance. This is in contrast to other studies highlighting the difficulty of persuading people with dementia to comply with public safety measures [3,51,73]. This may be related to the characteristics of the study sample. Participants showed no behavioral disturbances and were in the early stages of dementia, which might have positively influenced their compliance due to higher levels of mental capacity and thus understanding the reasoning behind the need for public health measures [74].

The lack or inadequacy of support options reported in this study was associated with the transition to remote use of services and social contacts. In Poland, older adults, including people with dementia, are often alienated from technological progress [75], which causes difficulties in benefitting from remote support provision [6,76], including social and medical care. To date, knowledge about and usage of telemedicine in Poland have been limited [77]. Testing this solution suddenly and widely during the lockdown has highlighted barriers to its successful implementation [49]. Apart from technical difficulties, one obstacle was the general reluctance of people with dementia to engage with remote contacts. As highlighted by other studies, people with dementia experience difficulties in coping with the lack of physical contacts and suffer its consequences [6,35]. However, informal carers were more proficient in digital media use and they could benefit from the possibilities offered by technology—both organizationally and emotionally [78,79,80]. This is encouraging when the introduction of technological solutions to dementia care is inevitable [81,82].

Although our study provides a broad sense of the various experiences of people with dementia and informal carers, some limitations need to be considered. More carers participated than people with dementia. This may limit the picture of the pandemic situation more towards the carer perspective. Moreover, the recruitment strategy enabled us to only interview people living in Wroclaw, which is among the largest cities in Poland, thereby excluding the experiences of people with dementia and carers residing in more rural areas, who are likely to be facing additional barriers in accessing care. Bearing these limitations in mind, future research needs to explore the experiences of people with dementia and informal carers in more rural settings. A further step would be to make international comparisons in order to explore how different health and social care systems have performed in the context of providing dementia care. It should be mentioned that there are also some limitations implicitly resulting from the adopted study design. Due to the specificity of qualitative research, at the expense of obtaining more in-depth, individual perspectives, the representativeness of the results is limited [38,44]. The presented outcomes can therefore be treated as a source for subsequent research questions further exploring the topics from the current study. Further qualitative research can, for example, involve focus group discussion on the issues identified but insufficiently elaborated in the analyzed interviews. Such an example could be remote communication in dementia care—a topic which, although present in earlier research [77,83], takes on a new meaning in the face of the COVID-19 pandemic [76,78,79,80]. Another issue worth exploring further in a quantitative design is coping strategies classified as a sub-theme of *psychological response.* A quantitative study may precisely characterize the focus of these strategies and their frequency.

## 5. Conclusions

This appears to be the first study focused on the experiences of people with dementia and informal carers related to the closure of social and medical services in Poland during the COVID-19 pandemic. The findings contribute to an emerging evidence base [1,6,36,51,62,71] on the urgent needs in dementia care, highlighting the importance of social health in dementia, pointing out the malicious effects of its deprivation and emphasizing the role of social support and services both during and beyond the pandemic. Considering the inadequate state of dementia care during the pandemic, and the early consequences already noted in people with dementia and informal carers, our findings indicate an urgent requirement to fortify and extend social support and medical services to provide improved care for those affected by dementia, and their informal carers, especially in the light of the ongoing pandemic. Despite vaccinations starting to be rolled out in certain countries, another pandemic wave is accelerating, and services need to adapt flexibly and urgently to prevent further rapid deterioration among this vulnerable group. These findings are also relevant to planning to mitigate the impact of other future pandemics, not just COVID-19.

## Figures and Tables

**Figure 1 healthcare-09-01677-f001:**
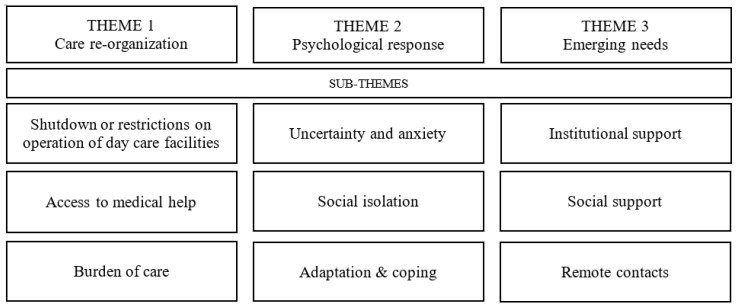
Themes and sub-themes.

**Table 1 healthcare-09-01677-t001:** Demographic characteristics of people with dementia and informal carers.

*N* (%)	People with Dementia (*n* = 5)	Informal Carers (*n* = 21)	Total Sample (*n* = 26)
Gender			
Female	4 (80%)	13 (61.9%)	17 (65.4%)
Male	1 (20%)	8 (38.1%)	9 (34.6%)
Relationship with PLWD			
Spouse	-	6 (28.6%)	-
Adult child	-	15 (71.4%)	-
Living with PLWD			
Yes	-	13 (61.9%)	-
No	-	8 (38.1%)	-
Dementia subtype			
Alzheimer’s disease	2 (40%)	8 (38.1%)	10 (38.5%)
Mixed dementia	2 (40%)	6 (28.6%)	8 (30.8%)
Vascular dementia	0 (0%)	4 (19%)	4 (15.4%)
Not specified	1 (20%)	2 (9.5%)	3 (11.5%)
Dementia in Parkinson’s disease	0 (0%)	1 (4.8%)	1 (3.8%)
Mean (SD), [Range]			
Age	78 (+/−6.6) (70–87)	* 81.5 (+/−4.7) (75–85) ** 63.1 (+/−9.9) (52–80)	80.8 (+/−5.14) (70–87)
Years of education	10.8 (+/−1.64) (9–12)	12.9 (+/−2.92) (9–17)	12.5 (+/−2.82) (9–17)

* Age of people with dementia living with/cared for by informal carers; ** age of informal carers.

## Data Availability

The data analyzed during this study are included in this published article and its Supplementary Information files. The full transcripts of the interviews analyzed in the study are available in Polish from the corresponding author on reasonable request.

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
