# Peer review of "The Experiences of People with Dementia and Informal Carers Related to the Closure of Social and Medical Services in Poland during the COVID-19 Pandemic—A Qualitative Study"

_healthcare, 2021, doi:10.3390/healthcare9121677_

Round 1
Reviewer 1 Report
This is a manuscript with great potential. The introduction provides useful and timely information, the structure of the manuscript is internally coherent and the conclusions are original. The quality of the presentation is very good and it is evident that much time has been invested in this aspect. Nevertheless, the methodology followed and the small number of participants make the generalization of the results and the replicability of the study difficult. While this is a topic that will be of interest to readers, the methodology and results need to be treated in a way that allows the study to be replicated. To this end, the qualitative information needs to be converted into quantitative information that can be studied using inferential statistical indices. Also, the number of participants needs to be increased considerably. Thank you for your attention.
Author Response
Thank you for your positive feedback on many aspects of the structure of our manuscript. We are also grateful for your critical comments, which, we believe, helped us to improve some sections of our manuscript and make it more methodologically transparent. In relation to your comment regarding the representativeness of the sample, we strengthened the Method section with the broader characteristics of the specificity of the study design we adopted (lines 149-154). We added more details about the data treatment (lines 188-205) to make a data analysis procedure more clear-cut for the readers, enabling the replication of our study. We also provided more extensive description of the limitations inevitable in a qualitative design in the Discussion with some additional implications for future research (lines 472-484). We argue that the quantitative design can be applied to the further studies, based on the results of the current research. We hope that the changes introduced to the text will allow for a positive assessment of the article.
Reviewer 2 Report
The article is very interesting due to the topic but the sample size is very small and the results are not enough representative. It is recommended to wide the explanation about the treatment of data and to enlenght the research with other methods as discussion groups or so.
Author Response
Thank you for your succinct and valuable review. Adopting the research design, we were aware that we were giving up good representativeness in favor of a deeper insight into individual experiences. However, your comment prompted us to review the discussion again and see if we are drawing overly generalized conclusions. As suggested, we strengthened the Method section with a broader description of data treatment (lines 188-205). We hope that it will enable the readers to better understand the specificity of the qualitative design we adopted and will clarify how the data were processed to obtain final qualitative themes and sub-themes. We would like to thank the Reviewer for pointing out a very interesting direction for expanding research with the focus groups and other research methods. We have included this remark in the section on the directions for further research (lines 472-484). In the present study, we have drawn a primary picture of the effects of a COVID-19 pandemic on individuals. These initial results can form a basis for further focus group discussion methodology or quantitative research where specific outcomes form the current study (e.g. coping strategies in adaptation to the pandemic situation) can be more widely investigated. We hope that the introduced changes address the issues raised by you in a satisfactory manner and allow for a positive assessment.
Reviewer 3 Report
Thanks for recommending me as a reviewer. The purpose of this study was to investigate the experiences of people with dementia and informal caregivers during the COVID-19 pandemic, while available social and medical services were closed in Poland. Qualitative research on the topic of the COVID-19 pandemic is meaningful. If the authors complete the revision, the quality of the study will be further improved.
- The introduction section is well written.Over the past two years, many previous studies have been accumulated to identify the burden of care and related factors due to the COVID-19 pandemic. If authors describe the burden of care and related research trends by referring to more previous studies in the introduction section, it can help readers understand.
2. line 90-92: If authors specifically describe the type of qualitative research in the Methods section, it may help readers understand.
3. The research results are well written.
4. In Table 1, are "PEOPLE WITH DEMENTIA (n=5)" and "INFORMAL CARERS (n=21)" necessarily capitalized?
5. If the authors add implications for future research in the discussion section, it may be helpful to the reader.
Author Response
Thank you for your valuable and substantive review that indicated us very precisely how to improve our manuscript.
As has been suggested in the comment no. 1, we expanded the Introduction section with some examples of pre-pandemic studies on informal carers’ burden. We were aiming to briefly illustrate how the results from pre-pandemic studies on psychological, social and medical functioning of informal carers correspond to the current studies, conducted during the COVID-19 pandemic (lines 112-118).
Regarding the comment no. 2 we went into more details about the qualitative study design and provided some arguments why we adopted this model in the current research (lines 149-154). Further, we provided more precise description of the data treatment to make the Method section and the research design used more transparent for the readers (lines 188-205).
We have included the stylistic remark from point 4 and changed the capitalization (table 1).
Having regard to the comment no. 5, we broadened the Discussion of more limitations resulting from the adopted study design. Thus, we have derived additional conclusions on further research directions, pointing to proposals for research problems and possible research models (lines 472-484).
We are hopeful that introduced changes responds to the issues raised in your review and allows for a positive opinion.
Round 2
Reviewer 1 Report
Thank you for improving the manuscript and explicitly stating the limitations associated with the research design. I also advise you to think about the title, perhaps it is too long. Perhaps the initial article "the" could be eliminated.It is an interesting paper.
Reviewer 2 Report
Authors have made a remarkable effort to fit all the suggestions.